# On Overcompression in Continual Semantic Segmentation

## Abstract

Class-Incremental Semantic Segmentation (CISS) is an emerging challenge of Continual Learning (CL) in Computer Vision. In addition to the well-known issue of catastrophic forgetting, CISS suffers from the semantic drift of the background class, further increasing forgetting. Existing attempts aim to solve this using pseudo-labelling, knowledge distillation or model freezing. We argue and demonstrate that frozen or rigid models suffer from poor expressibility due to overcompression. We improve on these methods by focusing on the offline training process and the expressiveness of the learnt representations. Beyond the characterisation and demonstration of this issue in terms of the Information Bottleneck principle, we show the benefit of two practical measures: (1) using shared but wider convolution modules before final classifiers to improve scaling for new, continual tasks; (2) introducing dropout into the encoder-decoder architecture to improve regularisation and decrease the overcompression of information in the representation space. We improve the IoU on the 15-1 and 10-1 scenarios by over 2% and 3% respectively while maintaining a smaller memory and MAdds footprint. Last, we propose a new benchmark setting that lies closer to the nature of lifelong learning to drive the development of more realistic and valuable architectures in the future.

## 1 Introduction

Continual Learning (CL) aims to address the shortcoming of standard supervised learning that requires large quantities of labelled data. It pursues a more natural, human-like ability to quickly and continuously learn from small exposures to training data. The learner experiences a stream of tasks whose relatedness is not known beforehand [4] and has the potential to quickly learn a new task by re-using past experiences. Clearly, we have to prevent the model from naively storing all experiences. This constraint enforces the need to efficiently compress the knowledge to a modest size and limit the computation used to recreate representation for a task at hand. Recent work on Continual Learning focuses on the numerous ways to combat forgetting using replay methods, continually changing the models' architecture to adapt to new knowledge or distillation methods.

Class-Incremental Semantic Segmentation (CISS) is a recently recognised problem that, on top of forgetting, deals with the unique issue of *background shift* [2] of the unknown pixels (see Section 2.3.1). Existing work focuses on the efficient transfer of existing knowledge between the continual steps while acquiring enough information to work with the tasks at hand. The prevalent approach is using rigid encoders that aim to maintain initial features throughout training while focusing on attempts to learn new classes. As a result, the most successful approaches tend to fair well with initial classes while struggling with the ones learned in the online fashion.

Most problems, however, tend to also have an offline phase where the model is initialised and learns the initial data and tasks. Mirzadeh et al. [27] show that wide convolutional models outperform

standard ResNet-based encoders [17] in Continual Learning, proving that the choice of the initial architecture is fundamental to combating catastrophic forgetting. Further inspired by the Information Bottleneck principle [35] we claim that we should put just as much effort into the training of the *offline* phase as we put into the *online* phase for best results.

## 1.1 Deep neural networks and overcompression

When developing models for CL, it is important to develop from pre-trained models that have many diverse features. Adding additional filters to an existing model is resource intensive, and provides training challenges. Therefore, it is important to start with an extensive feature set that we can adapt to the different scenarios as they appear. However, in initial offline training on the limited data, the many unimportant features are pulled closer to existing relevant filters, and such excessive compression means that we might remove features that are crucial for learning a prediction of one of the future classes.

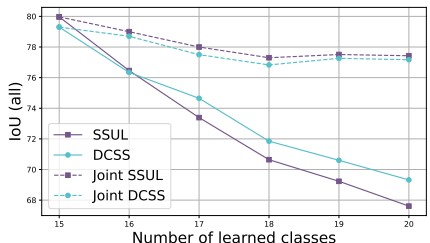

The Information Bottleneck principle (IB) [35] can partially explain and support the above difficulty of training Continual Learning models with gradient-based optimisation. The IB interpretation of learning suggests that hidden layers in neural networks learn to maximally compress the characteristics in the input data that are irrelevant to the task at hand, and where the higher level layers learn to compress first and the the lower level layers as training progresses. On the other hand, the uncertainty about the future requirements posed by CL requires us to maintain as many high-quality features as possible. Thus, it is in our best interest to maintain diverse features while limiting overcompression in Continual Learning.

Although the importance of the Information Bottleneck theory has been questioned by some papers [31], it serves as an interesting point of view on the learning phases of neural networks and can be

Figure 1: **IoU and Joint upper-bounds comparison** of the proposed DCSS model and SSUL [3] in the 15-1 scenario. DCSS achieves better online scores despite being worse in offline training, suggesting that we can improve CL performance by increasing the expressiveness of the learnt representations.

noticed empirically. We know that wider models better manage catastrophic forgetting [27], proving that passing information through the bottleneck, which inherently imposes compression, can have adverse effects on future tasks. Therefore, we use two simple approaches to verify if the reduction of overcompression can give us attainable gains in CL, in particular in CISS:

1. Increase the width of the last layer to reduce the bottleneck effect and increase the amount of information for each subsequent continual task;
2. Introduce dropout layer between the pre-trained encoder and decoder to introduce uncertainty into the model, encouraging a more robust representation that is later reused by the future steps.

By developing Dropout Continual Semantic Segmentation (DCSS), which incorporates only these two simple additions while improving on previous works by a sizeable margin (Figure 1), we demonstrate that improving the representation provided by the initial network has a significant impact on overall results.

## 2 Related work

### 2.1 Catastrophic forgetting

Catastrophic forgetting [30, 24] is a phenomenon observed primarily in Continual Learning where the earlier learned concepts are forgotten while incorporating the more recent samples. Forgetting appears when previously-learned representations are degraded by more recent exposures, a typical case in all SGD-based algorithms. Solutions proposed to address this issue can be grouped into the following categories:

**Rehearsal learning** In rehearsal learning, we utilise the fact that, although we have lost access to the original training data from the past, we are usually allowed to maintain the data in a different form. [3] store exemplar images in a small memory buffer that can be replayed to simulate previous data distribution. Compressed features in the form of embeddings [15, 20] can be used as an efficient form of memory that can also have advantages in terms of privacy. Finally, we can use generative strategies [32] that can efficiently store and reconstruct past experiences to enable the replay of raw images. [21] uses a brain-inspired dual-memory system where the new memories are consolidated from recent memory to a long-term memory using a generative model, similar to mechanisms that occur during sleep.

**Adaptive architecture** Other approaches focus on the adaptability of the model architectures. To integrate new classes and tasks, we can extend the architecture using feature extractors with domain-specific trainable layers [24, 39]. Model freezing can help with performance degradation during fine-tuning of the new models for new tasks. It is also possible to adapt the networks without explicitly adding new modules. [12, 11] dynamically re-arranges existing sub-networks, each specialised in one specific task, to account for new knowledge. Moreover, continually changing data distributions can be accounted for by explicitly correcting the classifier drift [1, 38].

**Knowledge distillation** Knowledge distillation methods consider the use of a *teacher-student* approach. A knowledgeable teacher model trained on past inaccessible experiences can inform the current model of the past. Therefore, distillation aims at constraining the model to prevent forgetting as it changes to adopt new data. There are several ways to constrain the model, with the most salient methods being applied to the weights [22], gradients [4, 25] or output probabilities [24, 29].

## 2.2 Information Bottleneck principle

Tishby et al. in the Information Bottleneck principle [35] claim that modern Deep Neural Networks undergo two phases of learning. During the generalisation phase, the model learns an internal feature representation to extract high-level information, used for final prediction. In the compression phase, unused and noisy features that prevent robust adaptation to the data are stripped away and removed from the feature space, ultimately improving useful features' signal quality. Tishby et al. measured the mutual information $I(X;T)$ and $I(T;Y)$, which quantify the hidden layer's information about the input and the output, where $X$ is the input, $Y$ is the output and $T$ is the internal representation. They showed that the amount of information about the output $I(T;Y)$ steadily increases until a high number of epochs is experienced, and the model starts overfitting. The information about the input $I(X;T)$ raises initially but quickly starts decreasing. This decrease can be considered as the compression phase because we remove information from the input that does not contribute much to the output. Compression accelerates rapidly when the number of training epochs is high.

## 2.3 Class-incremental semantic segmentation

### 2.3.1 Problem definition and notation

We follow the definition of van de Ven et al. [36] who use three distinct scenarios for Continual Learning: Task-Incremental, Domain-Incremental and Class-Incremental learning. This work focuses on the Class-Incremental setting initially formulated by Cermelli et al. [2] where we split the set of semantic segmentation classes into $t = 1, \ldots, T$ incremental tasks, with $t = 1$ being the offline step. The model learns each disjoint set of classes incrementally while trying not to forget the previous steps. At each step a new task $t$ arrives with a training dataset $D_t$ that consists of pairs ($\boldsymbol{x}^t$, $\boldsymbol{y}^t$), where $\boldsymbol{x}^t \in \mathcal{X}^N$ denotes an input image of $N$ pixels, and $\boldsymbol{y}^t \in (\mathcal{C}^t \cup \{c_b^t\})^N$ denotes the corresponding ground-truth pixel labels, where $\mathcal{C}^t$ is the set of classes that are seen in task $t$ and $c_b^t$ is the background class for that task.

At test time, the semantic segmentation model $f_{\boldsymbol{\theta}}^t$ is required to predict whether a pixel belongs to a class learned so far, $c \in \mathcal{C}^{1:t-1} \cup \mathcal{C}^t$, or the true background class $c_b$. However, the labels of an image from a task $t$ only contains classes from $\mathcal{C}^t$, not classes seen in past or future tasks. Thus, during training, the background label $c_b^t$ is also assigned to the pixels of potential objects that belong to past classes $\mathcal{C}^{1:t-1}$ and the future classes $\mathcal{C}^{t+1:T}$ (Equation 1).

$$c_b^t = c_b \cup \underbrace{\mathcal{C}^{\,1:t-1}}_{past} \cup \underbrace{\mathcal{C}^{\,t+1:T}}_{future} \tag{1}$$

This counter-intuitive behaviour of the background class' labels causes the problem of *background shift* [2], on top of the catastrophic forgetting found in CL. Each pixel with background label can be a class from the past, future or the actual background. This introduces unwanted noise into the model because the same object can be labelled differently, depending on the step $t$. Once the learning of task $t$ by the model $f_{\boldsymbol{\theta}}^t$ is done, the prediction for pixel $i$ of an input image $\boldsymbol{x}$ at test time is obtained by

$$\tilde{\boldsymbol{y}}_i^t = \underset{c \in \mathcal{Y}^{1:t}}{\arg\min} f_{\boldsymbol{\theta},c}^t(\boldsymbol{x}_i^t) \tag{2}$$

for all the classes learned so far in $\mathcal{C}^{1:t}$. The performance is measured by the Intersection-over-Union (IoU) metric for the three splits of $\mathcal{C}^{1:t}$. Cermelli et al. [2] introduce two different settings for Class-Incremental Semantic Segmentation: *Disjoint* and *Overlapped*. Both cases consider training examples where only the background class $c_b^t$ and classes in $\mathcal{C}^t$ are used as labels. In *Disjoint*, only the old and present classes can appear, while in *Overlapped* pixels can belong to any old, current and future classes. Thus, following the previous works, we also focus only on the Overlapped setting as it is more challenging and closer to real conditions.

### 2.3.2 Existing work

Class-Incremental Semantic Segmentation is a relatively new problem, with few works attempting to address it. Cermelli et al. in MiB [2] first defined the issue of *background shift*, a problem unique to Continual Semantic Segmentation. To alleviate the issue, they proposed to adapt the loss function such that it takes into consideration potential previous classes $\mathcal{C}^{1:t-1}$ that could be included in $c_b^t$. They sum the output probabilities of $\mathcal{C}^{1:t}$ and $c_b$ to match the available label $c_b^t$ that contains them when computing the cross-entropy loss. Conversely, they sum the probabilities of $\mathcal{C}^t$ and $c_b^{t-1}$ for knowledge distillation to match the output of $f_{\boldsymbol{\theta}}^{t-1}$. The summation of class probabilities to modify the cross-entropy and distillation losses is a novel approach in semantic segmentation, but it makes it hard to learn the underlying probability distribution for the classes at each pixel.

PLOP [9] extended the knowledge distillation on output probabilities from [2] with Localised Pooled Distillation [9], a form of heavy attention transfer that prevents from forgetting at the cost of decreased flexibility. Additionally for a task $t$, the target labels $\boldsymbol{y}_i^t$ are augmented with generated *pseudo-labels* [23] with a model $f_{\boldsymbol{\theta}}^{t-1}$ from a previous step. This yields the target semantic maps $\tilde{\boldsymbol{y}}^t$ containing background class $c_b^t$, pseudo labels $\hat{\boldsymbol{y}}^{t-1}$ and current labels $\boldsymbol{y}^t$.

SSUL [3] finds success with further increase of model's rigidity using model freezing. Instead of fine-tuning the whole model at task $t$, all parameters from $f_{\boldsymbol{\theta}}^{t-1}$ are being frozen in $f_{\boldsymbol{\theta}}^t$, preventing any changes to the prediction of classes $\mathcal{C}^{1:t}$. Moreover, SSUL uses saliency maps generated by a separate, off-the-shelf salient object detector [19] to predict a region of interest from the background which helps to differentiate the difference between true background and background that may contain a past or future class, which they label using the *unknown class*. Similarly to pseudo-labels, the saliency map is added to the labels $\boldsymbol{y}^t$ and predicted by the model during training. The unknown class $c_u$ and the background class $c_b$ predictions are combined at test time, representing the unclassified pixels. Taking advantage of the foreground prediction, SSUL performs weight transfer from the unknown class to the new classifier $\phi_{c_u}^{t-1} \to \phi_c^t$ at the beginning of each continual step.

## 3 DCSS model design

### 3.1 Wider classifiers with shared representation

Overcompression can hurt the learning of future tasks. Our goal is to maintain as many features as possible for future use. One simple way to achieve this is to use wider networks that are no longer a bottleneck to the information signal. To verify our hypothesis we decide to increase the number of channels in the last layer of the frozen, offline representation. In this way, we should have more information available for the continual steps.

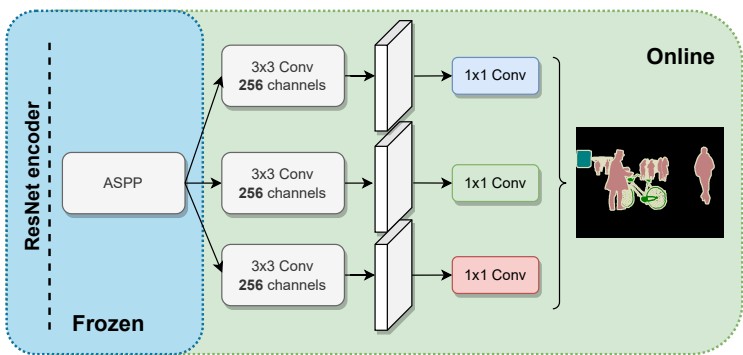

Figure 2: **High-level architecture of SSUL** [3]. Each step has its own $3 \times 3$ convolution block that is trained in the online phase.

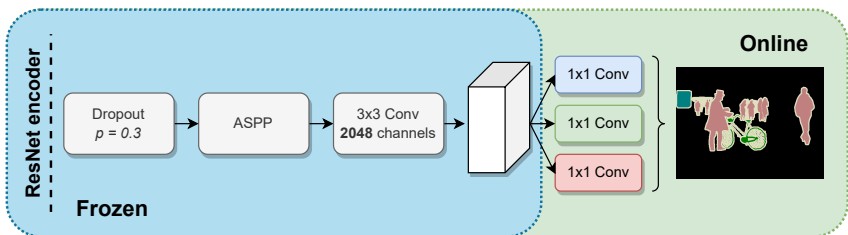

Figure 3: **High-level architecture of our DCSS** model. One HEAD module is trained in the offline step and frozen. Dropout layer is placed right after the pre-trained encoder to increase uncertainty in the learnt decoder.

In the DeepLabV3 architecture [5] used for this project the final $1 \times 1$ classifier is preceded by the ASPP module and one $3 \times 3$ convolution. The last $3 \times 3$ convolution is important to aggregate and mix multi-scale features produced by the ASPP module. We will refer to this $3 \times 3$ convolution layer as HEAD for readability reasons. We increase the size of HEAD output channels from 256, as in the original DeepLabV3, to 2048 channels. Hence, a more detailed feature space is available for the classifiers to exploit. Recent papers tackling CISS, including MiB, PLOP and SSUL, replicate the HEAD module for each step. Therefore, each set of classes $\mathcal{C}^t$ has its own HEAD and final classifier, as seen in Figure 2. However, the HEAD layer is a costly operation that has over 0.5M parameters with 256 channels and over 4.5M parameters when using 2048 channels. Therefore, to prevent the large parameter accumulation we use only a single HEAD layer, frozen after offline training and add only $1 \times 1$ convolutions at online steps $t > 1$. Additionally, we use depthwise-separable convolution in the shared HEAD to produce the same activation map using a cheaper operation [8] with fewer parameters and MAdds. In our work, we have used an intermediate width multiplier of 4 and 4 convolution groups. Standard shared HEAD has almost twice the parameters of the depthwise-separable implementation (4.71M vs 2.68M). The high-level architecture comparison can be found in Figures 2 and 3.

The reason why at step $t = 1$ the DCSS is larger than standard DeepLabV3, despite both being trained in an offline fashion, is the requirement of having a separate classifier for the background class $c_b$ and the unknown class $c_u$, which means that we already have three HEAD modules at $t = 1$, including the one for classes in $\mathcal{C}^{t=1}$. In contrast, increasing the size of the HEAD to produce 2048 channels with separable convolution as in DCSS, compared to 256 channels in SSUL, yields a significantly smaller model in the long run (Figure 4) while achieving better results than SSUL.

### 3.1.1 Dropout as a source of uncertainty

Dropout [34] randomly sets activations of a layer to $0$ with probability $p$, effectively disabling their contribution to the calculation of the output and introducing uncertainty that should hinder overcompression. Since the placement of the Dropout is essential, we follow Spilsbury et al. [33] and place the Dropout layer in between the encoder and decoder modules. The idea behind this decision

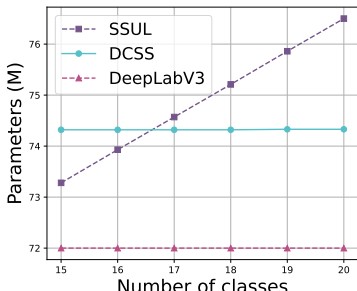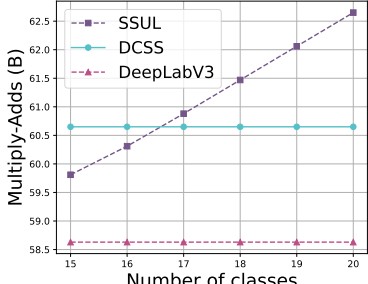

Figure 4: **Resource comparison** in the **15-1** scenario. SSUL adds significantly more trainable parameters at each step, surpassing DCSS size and complexity in just 2 steps.

is that we want to prevent interference in the pre-trained encoders while maximising the effect on the decoder.

We test two popular Dropout approaches in CNNs: standard Dropout and channel Dropout, where the whole convolutional channel can be dropped with probability $p$. Moreover, we use ScheduledDropout [33] that linearly increases the dropout probability from 0 to $p$ during training. During the exploration phase (generalisation), smaller values of $p$ help with feature accumulation. Heavier regularisation is more useful during the exploitation phase (compression), thus requiring higher values of $p$. In the end, we obtain similar benefits while improving learning convergence. Last, we suggest the removal of the Dropout after offline training. Limiting the access to features in online training will only restrain the model from using them while offering few benefits in generalisation since we freeze the encoder and are unlikely to learn new features.

## 4 Experiments

**Implementation details** The code for this project has been based on the implementations of PLOP [9] papers and SSUL [3] that use the DeepLabV3 architecture [5]. To compare the results of DCSS to previous work [2, 9, 3], we also use ResNet101 while keeping in mind that a smaller encoder would have a better performance-to-cost ratio. A standard learning rate of $0.01$ has been used, with the value decreased to $0.001$ for the backbone when using pre-trained weights [5]. We use *poly*[1] learning rate scheduler, as proposed by the DeepLabV3 authors. The same data augmentation has been used as in SSUL. We use a batch size of 24 for all experiments compared to the size of 32 used by SSUL, as we found more success with smaller values, especially for the continual steps. We use binary cross-entropy loss with sigmoid activation function, as in SSUL. Pseudo-labels are added to labels in the continual steps, with an entropy-based threshold of $\tau = 0.9$. In CISS problem we train each step for 50 epochs. The experiments were run on 4 RTX 2080Ti GPUs using the `torch.distributed` library [7] to work around the limited GPU memory. Each experiment, including the offline step, takes approximately 2-4 hours, depending on the scenario. In the case where the offline step can be reused, the training usually is decreased to 1-2 hours.

**PASCAL VOC 2012** The experiments are evaluated on the PASCAL VOC 2012 [10] semantic segmentation dataset with 20 foreground object classes and one common background class. The dataset is augmented with the extra contour annotations [14] of the PASCAL VOC 2011 dataset. Following [9], we split the training set and used 80% for training and 20% for validation, with the testing set left untouched for model evaluation. To help distinguish the background containing potential future classes from the true background, SSUL [3] proposed the use of an off-the-shelf salient object detector [19] to predict a region of interest. Thus, we follow their implementation and extend the labels of the PASCAL VOC training set to include the additional foreground class.

---

[1]The scheduler has been called "poly" by the authors of DeepLabV3, even though the learning rate is not a polynomial [37].

Table 1: **Main results for the CISS problem**. DCSS previous works in new classes (middle columns) in all scenarios and achieve improvements in combined scores in the more difficult scenarios with multiple tasks.

| Method | VOC 10-1 (11 tasks) | | | VOC 15-1 (6 tasks) | | | VOC 5-3 (6 tasks) | | | VOC 19-1 (2 tasks) | | | VOC 15-5 (2 tasks) | | |
|---|---|---|---|---|---|---|---|---|---|---|---|---|---|---|---|
| | 0-10 | 11-20 | all | 0-15 | 16-20 | all | 0-5 | 6-20 | all | 0-19 | 20 | all | 0-15 | 15-20 | all |
| ILT [26] | 7.15 | 3.67 | 5.50 | 8.75 | 7.99 | 8.56 | 22.51 | 31.66 | 29.04 | 67.75 | 10.88 | 65.05 | 67.08 | 39.23 | 60.45 |
| MiB [2] | 12.25 | 13.09 | 12.65 | 34.22 | 13.50 | 29.29 | 57.10 | 42.56 | 46.71 | 71.43 | 23.59 | 69.15 | 76.37 | 49.97 | 70.08 |
| PLOP [9] | 44.03 | 15.51 | 30.45 | 65.12 | 21.11 | 54.64 | 17.48 | 19.16 | 18.68 | 75.35 | 37.35 | 73.54 | 75.73 | 51.71 | 70.09 |
| SSUL [3] | 71.31 | 45.98 | 59.25 | 77.31 | 36.59 | 67.61 | **71.17** | 45.38 | 52.75 | **77.73** | 29.68 | **75.44** | 77.82 | 50.10 | 71.22 |
| DCSS (ours) | **73.34** | **50.20** | **62.32** | **77.66** | **42.69** | **69.33** | 68.10 | **48.83** | **54.34** | 77.22 | **36.85** | 75.30 | **77.49** | **51.49** | **71.30** |
| Joint (V3) | 78.41 | 76.35 | 77.43 | 79.77 | 72.35 | 77.43 | 76.91 | 77.63 | 77.43 | 77.51 | 77.04 | 77.43 | 79.77 | 72.35 | 77.43 |
| Joint (DCSS) | 77.80 | 76.58 | 77.22 | 78.77 | 72.25 | 77.22 | 76.29 | 77.61 | 77.22 | 77.30 | 75.60 | 77.22 | 78.77 | 72.25 | 77.22 |

**Evaluation metrics** The difficulty of the Class-Incremental challenge also depends on the number of steps and the number of new classes. Thus, our model is evaluated on five different scenarios generated from the PASCAL VOC, each with a varying level of difficulty: **10-1** (11 tasks), **15-1** (6 tasks), **5-3** (6 tasks), **19-1** (2 tasks) and **15-5** (2 tasks). The numbers in each scenario define the number of classes introduced at each step. For example, **5-3** (6 tasks) means learning 5 base classes at the offline step $t = 1$, followed by 5 incremental steps $t \in \{2, \ldots, 6\}$ introducing 3 new classes at each step, yielding 6 training steps covering all 20 classes for PASCAL VOC. We report the results as mean intersection-over-union (IoU) for three categories of classes: old (offline), new (online) and all classes combined (offline + online). This split helps to distinguish the difference between forgetting and issue with obtaining the knowledge for the new tasks.

## 4.1 Experimental results

In Table 1 we observe that DCSS consistently outperforms other models while having a simpler and more extendable architecture that is also easier to train (Figure 3). In the most popular **15-1** scenario, DCSS achieves a 1.72% improvement over SSUL. The biggest overall gain of over 3% can be found in **10-1** scenario that has a larger number of continual steps. Most importantly, in all scenarios we have improved the mean score of the new, continual classes, up to 6% in the case of **15-1**. This gain can be a surprising result, considering that we are only training a single 2048-dimensional vector at each step. We conclude that the current counter-forgetting measures used in CISS effectively prevent models from adding new features to the representation and, thus, it is enough to simply learn a linear mapping of *existing* features during the continual phase.

To compare our results to the ones produced by previous works, we have to constrain ourselves to a model of a similar learning capacity. Figure 1 proves that we increase Continual Learning performance despite the decreased overall performance of DCSS in offline learning, further signifying the importance of our contributions to online learning. What is most important, all of this is achieved with a smaller and easier to train model. In **15-1** DCSS has over 2M parameters less than SSUL (Figure 4), while in the extreme case of **10-1** the model is actually smaller by over 5M parameters and the difference grows linearly with each additional step.

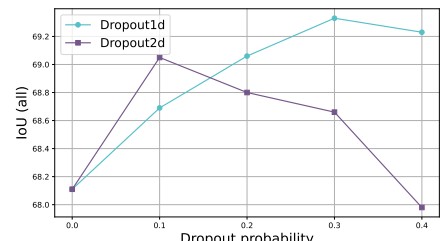

Figure 5: **IoU comparison of Dropout** in **15-1** scenario. We notice that a small amount of uncertainty produced by the Dropout can increase the IoU of the model.

Figure 5 shows improvement over the range of the Dropout probabilities $p$ compared to no Dropout ($p = 0$). Simply by adding the Dropout layer during the offline training we have increased the performance during the online steps, yielding a more accurate model on average. We have found more success with the standard dropout (Dropout1d) rather than the channel dropout (Dropout2d), despite literature stating the reverse for CNNs [33, 18]. The results follow our intuition, which suggests that dropping the whole channels removes a feature across the whole image, whereas

Table 2: **Experimental results on the proposed RCISS protocol**. Both SSUL and DCSS decrease their performance in new tasks, although the scores lie surprisingly close to the original scores. This suggests that more training cannot extract missing features from the frozen model.

| Method | Epochs | VOC 10-1 (11 tasks) | | | VOC 15-1 (6 tasks) | | | VOC 5-3 (6 tasks) | | | VOC 15-5 (2 tasks) | | |
| | | 0-10 | 11-20 | all | 0-15 | 16-20 | all | 0-5 | 6-20 | all | 0-15 | 15-20 | all |
|---|---|---|---|---|---|---|---|---|---|---|---|---|---|
| SSUL [3] | 50 | 71.31 | 45.98 | 59.25 | 77.31 | 36.59 | 67.61 | 71.17 | 45.38 | 52.75 | 77.82 | 50.10 | 71.22 |
| DCSS | 50 | 73.34 | 50.20 | 62.32 | 77.66 | 42.69 | 69.33 | 68.10 | 48.83 | 54.34 | 77.49 | 51.49 | 71.30 |
| SSUL [3] | 1 | 73.23 | 38.10 | 56.50 | **77.34** | 32.44 | 66.65 | 69.90 | 27.52 | 39.63 | 76.68 | **47.64** | 69.76 |
| DCSS | 1 | **73.78** | **43.26** | **59.24** | 77.16 | **37.46** | **67.71** | **68.44** | **38.84** | **47.30** | **77.46** | 45.60 | **69.88** |

standard dropout can work inter-channel, which is vital in pixel-level tasks. Removing whole channels can prevent the propagation of features, while removing specific neurons can decrease the reliance on more salient parts of the image and promote more holistic attention.

### 4.1.1 New protocols and evaluation

We have seen the performance of DCSS on the tasks proposed initially by [2]. DCSS offers a better IoU score while being a generally cheaper model to train in the long run. We argue, however, that the evaluation protocol has an unrealistic assumption about the ability to store the continual training data and use it multiple times over the step $t$. This is an unnatural setting that promotes irrelevant models. Continual Learning models should aim to efficiently use the available data and aim for an approach similar to few-shot learning, where only a few examples for each class are available. Therefore, we also evaluate our DCSS model on the proposed Restricted Class-Incremental Semantic Segmentation protocol (RCISS), where we train each continual step for only one epoch instead of the 50 as in SSUL. This scenario lies closer to the *lifelong learning* setting by Chaudhry et al. [4].

In the experiments on RCISS we remove learning rate scheduling and use a fix learning rate of $0.01$. Moreover, we reduce the batch size in the continual phase from 24 to 4, effectively increasing the number of backpropagation steps in the hope that, while being noisy, it will better utilise the limited exposure to data. Table 2 shows the result of our experiments on four scenarios in RCISS, similar to previous scenarios in standard CISS. We notice that both models struggle more with learning multiple classes at once (**5-3**), although DCSS manages to outperform SSUL in the continual classes by a considerable margin. However, we notice that the results for 1 epoch are still similar to the ones with full 50 epochs of training. We find this result interesting as it signifies the difficulty of extracting performance in continual learning even with extensive training.

## 5 Discussion and conclusions

We showed that the current approaches to CISS problem put too much importance on the continual phase of learning. The small difference between RCISS and CISS shows that there is a limit to the amount of information that can be obtained by simply learning new tasks with a continual manner while having significant constraints on the model flexibility to prevent catastrophic forgetting. The balance between rigidity and flexibility is hard to achieve, and most recent works found success with the focus on the former. With heavily constrained or frozen representations we are unable to adapt to new tasks or classes. Thus, assuming that the potential cost of adding features is much higher than removing them, we take a step back to consider what can be done to maintain potentially useful features for the future that would be pruned otherwise. We show that with a few simple changes to the offline training protocol we can propagate more information to future learning steps, lessening the need for complex online methods.

We adapted the idea of wide continual networks [27] for model freezing. Wider architecture of the decoder can help to maintain more features for CL; surprisingly, even though we are essentially learning a linear mapping of the embedding from the frozen model we improve the IoU in CISS. This further suggests that having a rich representation outweighs most attempts at learning new features using additional trainable layers while having the benefit of being easily scalable and far simpler to train. The simple addition of a single Dropout layer further improved the IoU, surpassing all current approaches based on the DeepLabV3 model.

Despite that, our work aims to show not the benefits of the proposed methods but rather the general lack of consideration of overcompression in Continual Learning. Information Bottleneck principle suggests that deep learning models have a generalisation and compression phase. Although useful in offline learning, compression is counterproductive in Continual Learning where we don't know the future requirements. We believe that most short-term gains in CL could be achieved by focusing on training with future's uncertainty in mind. Further gains could be achieved with a different approach to learning that will increase the generalisation. Recent trends tend to remove the reliance on supervised training for a more contextualised signal using self-supervised learning in the form of contrastive models [16, 6] or multi-modal representation learning [13], where a more holistic type of learning must appear to capture the whole context. Altogether, these approaches should allow for encoders that are less task-specific, providing the required flexibility in Continual Learning.

## 5.1 Future work

For subsequent work on the topic of CISS, we think a good idea would be to look at expanding our findings to different architectures. For example, SATS [28], used a Transformer-based encoder and self-attention transfer to achieve state-of-the-art results on CISS. So, while we expect our encoder-agnostic findings to maintain their validity for other architectures, it would be interesting to explore the topic of overcompression of Transformer models in the context of CL. Additionally, the research community is in dire need of dedicated Continual Semantic Segmentation datasets, which need to emphasize the issue of the background shift problem and introduce temporal and spatial locality of the data, which should spur new interest in this topic.

## Societal impacts

This paper touches upon the topic of Continual Learning which greatly increases the move towards more private and efficient training. Our contribution emphasizes the need for optimal offline training procedures that have immense impact on the future utility during continual phase, improving the ease of training and, in turn, decreasing the required compute and data. We do not see any foreseeable negative impacts of this work while noticing the positive impacts of more efficient CL models that should reduce the computational burden in the short term and allow for more intelligent agents in the long term.

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
