# OpenReview forum: "On Overcompression in Continual Semantic Segmentation"
_NeurIPS.cc/2022/Conference — NeurIPS 2022 Submitted_

### Official Review · Reviewer_tdRz · 2022-07-05

**Rating:** 3
**Confidence:** 3
**Soundness:** 2 fair
**Presentation:** 2 fair
**Contribution:** 1 poor

**Summary:**

In this work, the authors state that overcompression is a vital problem for Continual Learning and introduce two simple improvements for Class-Incremental Semantic Segmentation (CISS) problem. The first improvement is widening the network layers before the final classification layer to improve feature quality. The second improvement is adding a dropout layer after the encoder during the offline phase of the model training to create more robust features for further CL steps.

**Questions:**

1) lines 208-210: 'The idea behind this decision is that we want to prevent interference in the pre-trained encoders while maximising the effect on the
decoder.' Explain in detail this sentence, please. What has interference prevention in common with a frozen encoder, and what effect should be maximized on the decoder?
2) Why is DCSS worse in offline training (Fig. 1) while having more parameters?
3) What are the SD of results presented in tables 1-2?
4) How does the proposed evaluation scheme differ from a standard online learning setting, where the model is allowed to see each data sample once?
5) What are the limitations of the proposed improvements?

**Limitations:**

Limitations of the proposed improvements were not described. The authors mentioned general positive societal impact of efficient CL training.


**Strengths And Weaknesses:**

Strengths:
The authors focus on an important topic of overcompression, which has a negative impact on continual learning model training. Also, They show an increase in performance in specific setups using two proposed improvements. A possible novelty is using dropout layer only during the offline training phase of CL model to prevent overcompression.

Weaknesses:
Originality:
The paper lacks originality. The authors propose to use previously introduced improvements for a particular problem setting. The proposed 'new' evaluation framework is also not original. Likely it is a kind of a standard online learning setup.

Quality:
One of the main claims (line 8, lines 78-80) that 'we can improve CL performance by increasing the feature expressiveness of the learnt representations' is not directly supported with evidence within the work. The authors do not use any metrics to show improvement in the representations. Without additional ablation study, it is not clear
how parts of the solution impact the overall results.

There are multiple minor improvements or ideas that are not strongly connected. For example, the evaluation scheme proposal is not directly connected to other presented ideas. The authors state that they did not have enough computing resources to provide error bars for reported results.

Clarity:
The work is not self-contained. For example, the authors mention the background shift problem multiple times. They state in line 153 that this is a unique problem of Continual Semantic Segmentation, but they do not explain the essence of this problem. It is unclear how different parts of the related work section relate to the proposed solution.

The results in bold in Table 2 are not the best in columns. This is misleading. Some words are missing in the Table 1 caption.

---

> ### Comment · Reviewer_tdRz · 2022-08-08
> **No rebuttal**
>
> The authors have not responded to my concerns, therefore I stick to my initial score.

---

### Official Review · Reviewer_vqZ3 · 2022-07-09

**Rating:** 4
**Confidence:** 3
**Soundness:** 3 good
**Presentation:** 3 good
**Contribution:** 2 fair

**Summary:**

This paper proposes to tackle continual semantic segmentation by addressing the overcompression issue in model learning. In order to improve the generalization ability of representation to upcoming tasks, the authors draw inspiration from previous studies in the field and introduce wider convolution for final feature extraction as well as apply dropout to the output of the feature encoder.  The proposed method achieves good performance on the Pascal VOC dataset.


**Questions:**

- How about placing the Dropout and the wider convolution layer in different locations of the network?
- How does the proposed method perform under the "Disjoint" setting?

**Strengths And Weaknesses:**

Pros:
+ This paper is clear and easy to read.
+ The proposed method is simple and efficient.
+ The proposed method achieves good performance on PASCAL VOC dataset.

Cons:
- The contribution of this paper is limited. Previous works have demonstrated the effectiveness of wide convolution layers[a] and Dropout[b] for continual learning problems. This paper basically applies the techniques to the semantic segmentation task, which can be hardly considered as a significant contribution.

- The authors try to interrupt the effectiveness with the Information Bottleneck principle, yet either theoretical or empirical explanations are missing. Only the improvements in final accuracy are not sufficient to convince the reader about the paper's completeness. By the way, the term "generalisation phase" sounds wired for explaining the Information Bottleneck principle in Sec.2.2.

- Experiments are only conducted on the Pascal VOC dataset, while comparisons on ADE20K are missing. Since the proposed method is designed with a fixed feature extractor, it looks hard to generalize in situations the base class set is small and the new task set is large.

In summary, this paper presents a simple framework for continual semantic segmentation. Yet the contributions are insignificant, and the experiments are insufficient.

[a] Mirzadeh et. al., Architecture Matters in Continual Learning.
[b] Mirzadeh et. al., Dropout as an Implicit Gating Mechanism For Continual Learning.

---

### Official Review · Reviewer_TqWg · 2022-07-11

**Rating:** 3
**Confidence:** 4
**Soundness:** 2 fair
**Presentation:** 2 fair
**Contribution:** 2 fair

**Summary:**

The authors firstly reveal the overcompression of features causes model degradation when training the backbone of the continual segmentation model. They try to explain the reason from the aspect of the information bottleneck principle. Then they retrain the backbone combined with the extra Dropout layer and train the classifier with a greater number of features than SSUL. The experimental results show the effectiveness of their proposed methods.

**Questions:**

1. As the statement, high-quality features are required to solve the CL tasks related to the model structure. How to explain or define the high-quality or low-quality features? Why can a wider network extract the higher-quality features?
2. The number of output channels is different from the original Deeplab V3, what datasets use to pre-train the model? How to pre-train the backbone model? Is it initialized with the DeepLabV3 pre-trained parameters?
3. A wider network structure largely increases the number of model parameters which may enhance the model memory and manage the catastrophic forgetting problem. Obviously, the increasing number of channels means a higher capability of the model. More channels mean more features but do not mean the overcompression problem can be tackled. So, why can the increasing number of output channels solve the overcompression model? The ablation study does not compare the result from SSUL head modules with the same number of channels. If SSUL model has more channels, what will happen?
4. During the model design phase, most models adopt dropout layers to enhance the feature extraction. So as your model, why can dropout layers solve the overcompression problem?  What is the difference between the dropout layer from your and others? As your mentioned, all layers suffer from the overcompression problem. Why is the dropout layer only inserted before the ASPP module? What if the dropout layer inserts in all the convolutional layers?


**Limitations:**

Prior knowledge about the SSUL model needs to be listed in more detail. Some experimental settings should be clearly described especially your proposed module or methods.
For Question 3, conduct some ablation study in exploring the same number of channels in the SSUL Head layer.
For Question 4, conduct experiments on some dropout layers add after the other convolutional layers.
Experimental results partially show the effectiveness of your proposed methods. Try to explain in theory.


**Strengths And Weaknesses:**

This proposed method is based on SSUL but the introduction about SSUL is not detailed enough which may lead to confusion and misunderstanding. From the whole paper, it seems that a strong backbone can significantly improve the accuracy of CISS while the classifier just needs to classify features. The authors reveal the overcompression of features cause model degradation problem when training the backbone and some tricks for alleviating the problems. While their proposed methods try to extract more so-called robust or high-quality features for prediction, these methods can not solve the overcompression problem. Experimental results partially prove the effectiveness but lack theoretical proof and ablation study.

---

> ### Comment · Reviewer_TqWg · 2022-08-10
> **No rebuttal**
>
> The author did not submit a response to my concerns, so I keep the original score.

---

### Official Review · Reviewer_Wub6 · 2022-07-12

**Rating:** 4
**Confidence:** 4
**Soundness:** 2 fair
**Presentation:** 3 good
**Contribution:** 2 fair

**Summary:**

- The paper considers the problem of continual learning on semantic segmentation.

- The authors propose two simple ideas to improve the architecture:
    - using shared but wider convolution modules
    - introducing dropout into the encoder-decoder architecture

- Experiments have been conducted on the PASCAL VOC dataset, outperforming previous approaches.

**Questions:**

- If the over compression is a problem,  what will a more powerful architecture will perform, as the authors pointed out in the future work.

**Ethics Review Area:**

["I don’t know"]

**Limitations:**

- limitation has been included in the paper.

**Strengths And Weaknesses:**

- strength
    - the paper is well-written, easy to follow.

- weakness
    - writing-wise, the paper discusses a lot on the information bottleneck, I don't know how relevant it is, as the further proposed solutions are more like empirical tricks.
    - I think the considered scenario does not reflect what happens in practise, ``Class-Incremental Semantic Segmentation: Disjoint and Overlapped. Both cases consider training examples where only the background class c_b^t and classes in C^t are used as labels.'' if we do have labels for task (t-1), why are we not allowed to use them ? if for efficiency, why can't use only a portion of them, like replay.
    - the proposed solutions, although simple, does not give too much insight, more like some tricks.
    - experiment on PASCAL is not good enough, if the idea is to continuously learn new task, or in fact learning to segment new categories, it would make sense to test at least on COCO, with 80 classes, or even LVIS, with over 1000.

Overall, I don't think this paper should be published on NeurIPS.

---

### Meta-Review · Area_Chair_3Vye · 2022-08-26

**Recommendation:** Reject
**Confidence:** Certain

**Metareview:**

This paper deals with continual learning in semantic segmentation.  Authors introduce wider convolution at final feature extraction layer and apply dropout to limit the overcompression issue.
No reviewer was convinced by the approach and they have raised many issues, including model design choice, training protocol and missing experiments.
No rebuttal has been provided by the authors.
As it is, this submission is not ready for publication, and we encourage the authors to consider the reviewers feedbacks for future publication.

**Award:**

No

---

### Decision · Program_Chairs · 2022-09-14

Reject